# Impact of annual community-directed treatment with ivermectin on the incidence of epilepsy in Mvolo, a two-year prospective study

Luís-Jorge Amaral [1,¤]*, Stephen Raimon Jada [2], Aimee Kemayou Ndjanfa[1], Jane Y. Carter[3], Gasim Abd-Elfarag[2,4,5], Samuel Okaro[2], Makoy Yibi Logora[6], Yak Yak Bol[6], Thomson Lakwo[7], Joseph N Siewe Fodjo[1], Robert Colebunders[1]

1 Global Health Institute, University of Antwerp, Antwerp, Belgium, 2 Amref Health Africa, South Sudan, 3 Amref Health Africa Headquarters, Nairobi, Kenya, 4 Access for Humanity, Juba, South Sudan, 5 School of Public Health, University of Juba, Juba, South Sudan, 6 Neglected Tropical Diseases Unit, Ministry of Health, Juba, South Sudan, 7 Vector Control Division, Ministry of Health, Kampala, Uganda

☉ These authors contributed equally to this work.
¤ Current address: Global Health Institute, University of Antwerp, Kinsbergen Centrum, Antwerp, Belgium
* luis-jorge.telesdemenesesdoamaral@uantwerpen.be

**Data Availability Statement:** The datasets generated during and/or analysed during the

## Abstract

### Objectives

The potential impact of cumulative community-directed treatment with ivermectin (CDTI) on epilepsy epidemiology in Mvolo County, South Sudan, an onchocerciasis-endemic area with high epilepsy prevalence, was investigated. Annual CDTI was introduced in 2002 in Mvolo, with interruptions in 2016 and 2020.

### Methods

Comprehensive house-to-house surveys in Mvolo (June 2020 and 2022) identified cases of epilepsy, including probable nodding syndrome (pNS). Community workers screened households in selected sites for suspected epilepsy, and medical doctors confirmed the diagnosis and determined the year of seizure onset. The incidence of epilepsy, including pNS, was analysed using 95% confidence intervals (CIs). Data on ivermectin intake and onchocerciasis-associated manifestations (itching and blindness) were collected.

### Results

The surveys covered 15,755 (2020) and 15,092 (2022) individuals, identifying 809 (5.2%, 95% CI: 4.8–5.5%) and 672 (4.5%, 95% CI: 4.1–4.8%) epilepsy cases, respectively. Each survey reported that a third of the surveyed population experienced skin itching, and 3% were blind. Epilepsy incidence per 100,000 person-years gradually declined, from 326.5 (95% CI: 266.8–399.1) in 2013–2015 to 96.6 (95% CI: 65.5–141.7) in 2019–2021. Similarly, pNS incidence per 100,000 person-years decreased from 151.7 (95% CI: 112.7–203.4) to 27.0 (95% CI: 12.5–55.5). Coverage of CDTI was suboptimal, reaching only 64.0% of

current study are freely available to other researchers. See supplementary information.

**Funding:** The study was funded by an R2HC grant of Amref South Sudan (Project ID: 40385) and the European Research Council (ERC 671055) to RC. L-JA received funding from the La Caixa Foundation (grant number B005782). JNSF received funding from the Research Foundation – Flanders (FWO, grant number 1296723N). The funders had no role in study design, data collection and analysis, decision to publish, or preparation of the manuscript.

**Competing interests:** The authors have declared that no competing interests exist.

participants in 2019 and falling to 24.1% in 2021 following an interruption in 2020 due to COVID-19 restrictions. Additionally, while 99.4% of cases had active epilepsy in 2022, less than a quarter of these had access to antiseizure medication.

## Conclusions

The observed decrease in epilepsy incidence despite suboptimal CDTI coverage highlights the potential impact of onchocerciasis control efforts and underscores the need to strengthen these efforts in Mvolo County and across South Sudan. As a proactive measure, Mvolo and neighbouring counties are transitioning to biannual CDTI. Furthermore, the substantial epilepsy treatment gap in Mvolo should be addressed.

### Author summary

In areas where many people have onchocerciasis, a parasitic infection spread by female blackflies, there is often a high occurrence of epilepsy. The latter includes nodding syndrome (NS), a type of epilepsy where the head repeatedly drops forward. This is the case in Mvolo County, South Sudan. Our study examined how the annual intake of ivermectin, a drug that prevents onchocerciasis, impacted the occurrence of new epilepsy cases, including NS, in the county. In June 2020, we visited selected sites and screened 15,480 individuals for epilepsy, with confirmation of diagnosis by a medical professional. This exercise was repeated in June 2022 with 15,092 individuals. We found fewer new epilepsy and probable NS cases after introducing annual ivermectin treatment. Still, not all eligible community members received the treatment, with 64% of participants reporting taking ivermectin in 2019 and only 24% in 2021. Our research further corroborates that ivermectin intake at the community level can help reduce the high number of epilepsy and NS cases in regions where onchocerciasis is found. Consequently, it is imperative that individuals at risk for onchocerciasis, particularly those within the 5–18-year-old age group, who are most susceptible to developing onchocerciasis-associated epilepsy, have access to annual ivermectin.

## 1. Introduction

Onchocerciasis, commonly known as "river blindness", is a neglected tropical disease caused by the filarial parasite *Onchocerca volvulus*. The infection is transmitted to humans through the bite of infective female blackflies (*Simulium* spp.) [1]. Blackflies breed in fast-flowing water, often surrounded by fertile land suitable for agriculture, the primary occupation in sub-Saharan Africa [2]. Clinical manifestations of onchocerciasis range from itchy skin, acute and chronic onchodermatitis and (palpable) onchocercal nodules to impaired vision and irreversible blindness [3]. Recent studies suggest a robust association between onchocerciasis and epilepsy, particularly in areas with sub-optimal onchocerciasis elimination programmes [4–7]. The onset of onchocerciasis-associated epilepsy (OAE) usually occurs in children between the ages of 3–18 years and includes nodding syndrome (NS) [8], which is characterised by atonic seizures of the neck, causing the head to "nod" forward with periods of reduced consciousness, frequently accompanied by cognitive impairment and stunted physical growth. Another OAE phenotype termed Nakalanga syndrome presents as delayed secondary sexual development,

often with morphological deformities and seizures [9]. While other possible causes have been hypothesised for NS, at the time of writing, onchocerciasis stands as the possible consistent risk factor [10,11].

Onchocerciasis-associated epilepsy has been particularly linked with *O. volvulus* microfilarial load, which denotes the number of microfilariae (the early larval stage of *O. volvulus* parasite) in the skin of the host [11]. As such, effective public health strategies aimed at preventing OAE should focus on decreasing the microfilarial load. The predominant intervention for onchocerciasis control and elimination is community-directed treatment with ivermectin (CDTI) [12]. Ivermectin is a microfilaricidal drug that eliminates microfilariae and temporarily sterilises female adult worms [13]. While alternative interventions, such as vector control, directly target the transmission of *O. volvulus*, their impact on the microfilarial load might be gradual [14,15]. This slower change is attributed to the ability of adult worms to produce microfilariae for over a decade, even in the absence of ongoing transmission [16], and the density-dependent processes that govern the lifecycle of the parasite [17].

South Sudan is known for having one of the highest levels of onchocerciasis endemicity in Africa [18]. The country recognised epilepsy, including NS, as a public health concern in onchocerciasis-endemic areas as early as 1997, especially in Western Equatoria State, with reports of high epilepsy prevalence in endemic areas dating back to 1946 [19,20]. As of 2014, onchocerciasis was prevalent in approximately half the country [18], affecting an estimated 4.31 million individuals in high-risk areas [18]. Retrospective studies have consistently demonstrated a significant decrease in the burden of epilepsy, including NS, following the implementation and strengthening of onchocerciasis elimination programmes in endemic communities [21–23]. By 2023, over nine million South Sudanese were estimated to live in areas requiring preventive chemotherapy for the disease through CDTI [24].

This study sought to retrospectively and prospectively investigate the incidence and prevalence trends of epilepsy, including NS, with the cumulative distribution of ivermectin in Mvolo County, South Sudan. The findings obtained were compared with those from two other foci with a high prevalence of epilepsy, including NS, and of onchocerciasis, namely Maridi County, South Sudan, and the Mahenge area, Tanzania, where more robust onchocerciasis control interventions have been implemented.

## 2. Methodology

### 2.1 Ethics statement

The study was conducted according to the World Medical Association Declaration of Helsinki, which outlines the ethical principles for medical research involving human subjects. Ethical approval was obtained from the Ministry of Health of South Sudan (MOH/ERB50/2019) and the University of Antwerp, Belgium (B300201940004). Written informed consent (signed or thumb-printed) was obtained from all participants before enrolment into the study, with parents or caregivers providing written consent for minors. Additionally, assent was obtained from children aged 7–17 years. All personal information was encoded and treated confidentially.

### 2.2 Study setting and population

The study was conducted in selected sites in Mvolo County (N6.060121, E29.952274) in Western Equatoria State, situated along the fast-flowing Naam River and home to an estimated population of over 65,000 individuals (Fig 1) [25]. Mvolo is predominantly inhabited by the Jur, a minority ethnic group in South Sudan whose main religion is animism. Although Jur is their mother tongue, local Arabic is a common language of communication. Geographically, within

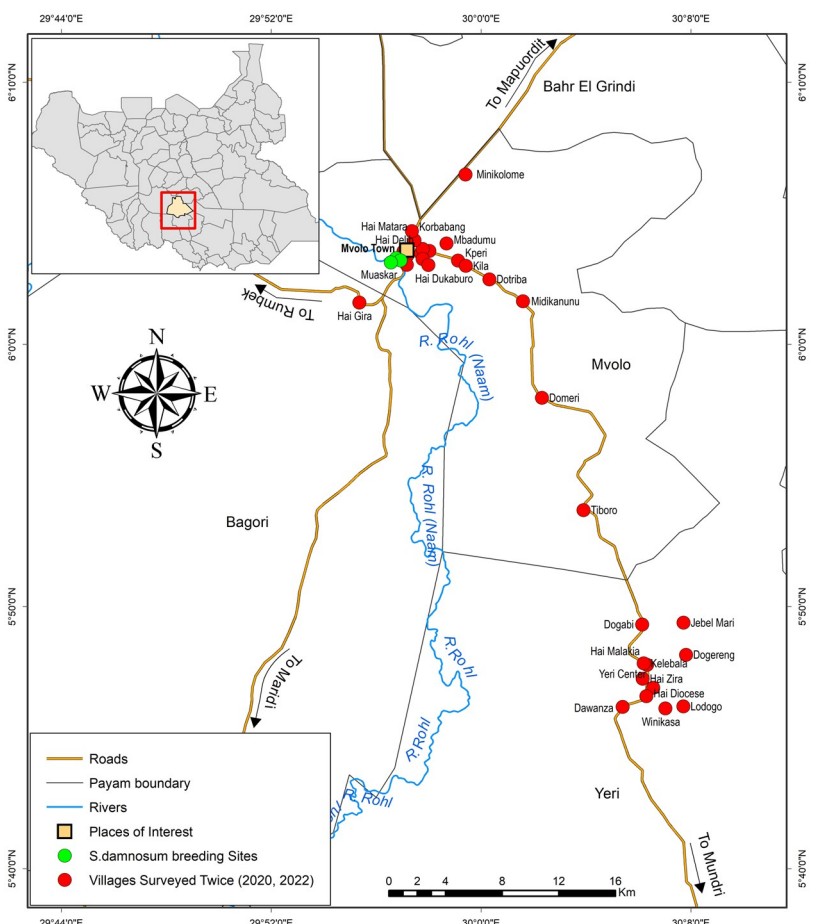

**Fig 1. Map of South Sudan and Mvolo County, including the Naam River, showing the breeding sites of blackflies (*Simulium damnosum*) and the sites visited during the house-to-house surveys (base map obtained from https://data.humdata.org/).**

Western Equatoria State, Mvolo is flanked by Maridi County to the southwest, Mundri West County to the south, and Mundri East County to the southeast. It neighbours Wulu and Yirol West Counties, Lakes State, to the north and Terekeka County, Central Equatoria State, to the east [25].

The Mvolo community primarily relies on farming and fishing for economic sustenance, taking advantage of the proximity to the Naam River, which is infested with several blackfly breeding sites. Animal husbandry is also practised as a supplementary livelihood activity, with cattle, goats and poultry being the primary livestock.

Mvolo is historically holoendemic for onchocerciasis, reporting over 90% of the population infected with *O. volvulus* and 10% blind in the 1960s [26]. This burden led to the relocation of the court and most of the population from Mvolo to *Simulium*-free areas [27]. In recent decades, the population of Mvolo has been growing, marked by a substantial number of births and children [19]. The CDTI programme in the county started in 2002, initially targeting a mere 12.4% of at-risk communities [28,29]. In 2006, the African Programme for Onchocerciasis Control (APOC) extended annual CDTI to the remaining communities [30], and CDTI has been sustained ever since except in 2016 and 2020 (the latter due to COVID-19 restrictions) [24,30–38]. By 2016, South Sudan aimed to eliminate *O. volvulus* transmission through annual

CDTI with 80% overall therapeutic coverage across all onchocerciasis-endemic regions [19], as per the 2016 World Health Organization guidelines [39]. Since then, the annual CDTI was delivered in September 2017, August 2018, October 2019 and August 2021.

### 2.3 Study design

This population-based study consisted of two repeated cross-sectional door-to-door surveys to identify and characterise individuals with epilepsy in Mvolo County and determine the year they experienced their first seizures (epilepsy onset) to estimate the annual incidence of epilepsy.

### 2.4 Study procedures

Two house-to-house surveys were conducted in Mvolo County over two weeks each, starting on 29 June 2020 and 4 June 2022, using a two-step diagnostic approach for epilepsy. First, trained research assistants recruited from the local community undertook door-to-door screening for epilepsy in households across selected sites. Upon obtaining informed consent, all household members provided their sociodemographic details and history of ivermectin intake during the most recent CDTI campaign (2019 for the initial survey and 2021 for the subsequent survey) and responded to a validated epilepsy screening questionnaire comprising five questions [40] (Questionnaire A in S1 Material). The main sociodemographic data recorded were age, sex, household ethnicity (only asked in 2020), household occupation, household residency status, household contact with animals (only asked in 2022), blindness and skin itching. If a family member was unavailable during the interview, information pertaining to them was sought from another family member present. If any of the aforementioned five epilepsy screening questions received a positive response, the participant was considered a suspected case of epilepsy and progressed to the second phase [40].

In the second phase, medical doctors trained to diagnose epilepsy either confirmed the epilepsy diagnosis or proposed an alternative based on an in-depth analysis of the medical histories and clinical examination of the suspected cases. This additional information from the "epilepsy confirmation questionnaire" (Questionnaire B in S1 Material) included clinical signs of onchocerciasis, namely blindness and onchocerciasis-associated skin lesions, as well as potential causes of epilepsy, cognitive impairment (orientation in time/space/person), frequency and types of seizures, Nakalanga features, level of disability, and prior intakes of ivermectin and antiseizure medication.

Questionnaires were translated into the local language (Arabic) by the local team during group discussions and subsequently back-translated into English to ensure no loss of meaning. Interviews were conducted orally. Trained research assistants administered the questionnaires for epilepsy screening and collected data on paper forms for the screening of epilepsy, while the clinicians confirmed or rejected the suspicion of epilepsy and documented additional information from persons with epilepsy into Open Data Kit software on tablet computers. Additionally, during household visits, a medical doctor (SRJ) oversaw the activities of the research assistants and other medical doctors and personally interviewed and assessed, together with six medical doctors, individuals with suspected epilepsy during each survey. More detailed research procedures have been previously reported [41].

### 2.5 Case definitions

A case of epilepsy was defined according to the International League against Epilepsy (ILAE) as an individual who experienced at least two unprovoked seizures a minimum of 24 hours

apart [42]. The types of seizures were organised according to the ILAE definitions set in 2017 [43].

A case of active epilepsy was identified as someone who reported experiencing at least one seizure within the past five years or is currently on antiseizure medication [44].

A probable case of NS was defined as a person with epilepsy characterised by repeated, involuntary episodes of head dropping during a brief period of reduced consciousness. Other potential features of NS included stunting, cognitive decline and spatial clustering of cases [8]. A person identified with probable NS experienced nodding seizures either at the time of the survey (ongoing nodding seizures) or had experienced them in the past.

A case of OAE was identified as an individual meeting a set of six criteria: (1) a history of at least two unprovoked epileptic seizures at least 24 hours apart; (2) residence in a known onchocerciasis-endemic region for at least three years; (3) inhabiting an area with a high prevalence of epilepsy and with families that have more than one child with epilepsy; (4) no obvious cause for the epilepsy; (5) onset of epilepsy between the ages of three and 18 years; (6) normal psychomotor development prior to the onset of epilepsy [8].

A potential "obvious cause of epilepsy", as referenced in the OAE definition, included specific events in the five years preceding the onset of epileptic seizures, such as perinatal anoxia, severe malaria, encephalitis, meningitis or a head injury leading to loss of consciousness.

A household or family was defined as all occupants of the same home who take meals together with or without parental ties. The household residency status was classified into native (household head born and bred in the village) or immigrant. Data for these variables were collected through the "household screening questionnaire".

Blindness was defined as the inability to discern the five fingers of a hand. Onchocerciasis-associated skin lesions were characterised by nodules, pruritic skin, leopard skin, scaly/thickened/wrinkled skin and severe itching. The participant's sense of orientation was used as a proxy of cognitive function; this was determined by assessing an individual's awareness of space, time and person during the study interview. Nakalanga features included the specific observations of visible growth retardation in individuals aged 16 years and above, often accompanied by thoracic or spinal abnormalities and the absence of external signs of secondary sexual development and "looks like a child" [9]. The level of disability was measured using the modified Rankin Scale, with categories as follows: no significant disability (scores 0 and 1); slight to moderate disability (scores 2 and 3); and moderately severe to severe disability (scores 4 and 5) [45]. This information was obtained from the "epilepsy confirmation questionnaire".

## 2.7 Data analysis

Continuous variables were inspected for normality using histograms and the Shapiro-Wilik test [46]. As they were not normally distributed, they were summarised using medians and interquartile ranges (IQRs). Categorical variables were presented as frequencies and percentages (%).

Crude estimates of the prevalence of lifetime epilepsy and probable NS were calculated by dividing the total number of confirmed cases by the total population screened in each survey. The prevalence results were reported as cases per 1,000 people screened with 95% confidence intervals CIs.

Overall epilepsy and probable NS incidence in the Mvolo population were estimated by taking the number of newly reported epilepsy and probable NS cases and dividing by the summed person-years of the at-risk population for each period under study: July 2013 to June 2015, July 2015 to June 2017, July 2017 to June 2019 and July 2019 to June 2021. These periods were chosen to observe the temporal trends of epilepsy incidence while taking into consideration the potential for recall bias and the possibility of missing newly developed epilepsy cases (Fig A in

S1 Material). For these calculations, the study assumed a stable population at risk for epilepsy during the years preceding each survey (2013–2019 for the initial survey and 2019–2021 for the follow-up survey). The incidence results were reported as cases per 100,000 person-years with 95% CIs.

Annual CDTI therapeutic coverage, defined as the percentage of the overall population that reported taking ivermectin, was calculated by dividing the number of individuals who reported receiving ivermectin in 2019 and 2021 by the total population surveyed each year. Additionally, the coverage was also calculated per age group. This methodology, relying on individual self-reports of ivermectin intake, is consistent with established methodologies employed in similar settings [14,47,48] and has demonstrated reliability as an indicator of treatment adherence in impact studies [49].

To account for the 33% overlap in surveyed populations in 2020 and 2022 (implying that the baseline and follow-up study populations were not completely independent), an adjusted Bootstrap method (producing 10,000 bootstrapped samples per comparison) was employed to calculate 95% confidence intervals bootstrap ($CI_b$) to assess differences between surveys [50–52]. This non-parametric approach generated bootstrapped samples incorporating the specific overlap proportion, representing individuals that were surveyed both once and twice. For dichotomous variables, bootstrapping estimated and compared proportions with $CI_b$. For continuous variables, bootstrapped medians and their $CI_b$ were used to evaluate median differences. A sensitivity analysis was performed comparing the $CI_b$ with those obtained using the Wilson score method with continuity correction. Additionally, all 95% CIs presented for single proportions and rates, including differences in ivermectin coverage, were determined using Wilson (w) Score Intervals ($CI_w$) with Yate's continuity correction for rare events [53]. A confidence interval that excluded the value of 0 was interpreted as statistically significant.

## 3. Results

### 3.1 Sociodemographic characteristics of households surveyed in 2020 and 2022

A similar number of households were surveyed in both surveys: 2,356 in 2020 and 2,402 in 2022. The median household size remained stable at 6 (IQR: 5–8) elements in both surveys (Table 1). Notably, there was an upward trend in the proportion of immigrant families in Mvolo as the latter increased by 4.0 (95% $CI_b$: 2.7–6.9) percentage points between 2020 and 2022, accompanied by an expected decrease in their median residence duration of 2 (95% $CI_b$: -3 –-1) years. The household ethnic distribution was only captured in 2020. Most families identified within the Jur ethnic group (92.1%), followed by the Muro (5.6%) and the Dinka (1.2%).

Distinct shifts in economic activities were observed between the surveys. While observing a slight decline, farming remained the predominant income source in Mvolo, with 93.6% of households relying on it in 2022. Concurrently, the surveys indicated a rise in the proportion of families whose members engaged in fishing (33.3% in 2022), craftsmanship (24.2% in 2022), and formal employment (20.9% in 2022). Other less common occupations reported were cattle rearing (14.8% of households in 2022) and pig rearing (0.9% in 2022). While there was a minor decrease in the proportion of families participating in activities at high risk for onchocerciasis (-3.2 percentage points, 95% $CI_b$: -3.8 –-1.7), a significant majority remained involved in these endeavours (94.8% in 2022).

Animal contact within households was documented exclusively in 2022. Most families reported interactions with poultry (80.4%) and goats (75.4%). In contrast, all other animals

**Table 1. Characteristics of households surveyed in 2020 and 2022.**

| Characteristics of households surveyed | | Survey | | Significant difference (95% CI_b) |
|---|---|---|---|---|
| | | **2020** | **2022** | |
| **Households surveyed** N | | 2,356 | 2,402 | - |
| **Household size** Median (IQR) individuals | | 6 (5–8) | 6 (5–8) | No (0–0 individuals) |
| **Immigrant households** N (%) | | 371 (15.8) | 476 (19.8) | Yes (2.7–6.9 pp) |
| **Length of residence of immigrant households in the current site** Median (IQR) years | | 9 (4–13) | 6 (4–13) | Yes (-3 –-1 years) |
| **Household ethnic distribution** N (%) | **Jur*** | 2,171 (92.2) | Not asked | - |
| | **Muro** | 132 (5.6) | | |
| | **Dinka** | 29 (1.2) | | |
| | **Other/unidentified**** | 24 (1.) | | |
| **Household economic activities** N (%) | **Farming** | 2,292 (97.3) | 2,248 (93.6) | Yes (-5.5 –-3.0 pp) |
| | **Fishing** | 698 (29.6) | 800 (33.3) | Yes (2.3–7.7 pp) |
| | **Formal employment** | 248 (10.5) | 501 (20.9) | Yes (6.7–10.7 pp) |
| | **Craftsmanship** | 177 (7.5) | 582 (24.2) | Yes (13.1–17.2 pp) |
| | **Cattle rearing** | 382 (16.2) | 356 (14.8) | No (-2.1–2.0 pp) |
| | **Pig keeping** | 17 (0.7) | 21 (0.9) | No (-0.2–0.8 pp) |
| | **Onchocerciasis high-risk activities: farming and/or fishing** | 2,308 (98.0) | 2,278 (94.8) | Yes (-3.8 –-1.7 pp) |
| **Household contact with animals** N (%) | **Poultry** | Not asked | 1,930 (80.4) | - |
| | **Dogs** | | 816 (34.0) | |
| | **Cats** | | 510 (21.2) | |
| | **Goats** | | 1810 (75.4) | |
| | **Cattle** | | 701 (29.2) | |
| | **Sheep** | | 533 (22.2) | |
| | **Contact with at least one of the above animals** | | 2,286 (95.2) | |

CI_b–Bootstrap confidence interval; IQR–Interquartile range; N–Number; pp–Percentage points.

* Sub-ethnicities included in the Jur ethnic group were Jur Modo/Beli (2,041), Nyamusa/Nyamusa Molo/Kpelikiri (74), Wira (52), Lali (2), Lori (1) and Sopi (1).

** Such as Madi (4), Baya (3), Paliki (2) and Nuba (1).

interacted with less than half of the households surveyed. Cumulatively, 95.2% of families had contact with at least one of the animal species reported.

## 3.2 Sociodemographic characteristics of individuals surveyed in 2020 and 2022

The sex distribution was consistent across both surveys ($p > 0.05$). However, there was a discernible shift of 2 years (95% CI_b: 1–2 years) towards a younger population (Table 2). This trend was consistent between immigrant and native households. Most individuals were available for household interviews in both surveys, with a marginal increase from 83.3% in 2020 to 85.9% in 2022.

Regarding clinical manifestations potentially related to onchocerciasis, the reported prevalence of blindness remained consistent across surveys, at 2.8% in 2020 and 2.9% in 2022. The median age of blind individuals decreased from 2020 to 2022 by 4 (95% CI_b: 3–10) years, possibly reflecting the younger demographics. Additionally, there was a slight increase in the proportion of people reporting skin itching, which stood at 35.1% in 2022 from 32.2% in 2020.

**Table 2. Characteristics of individuals surveyed in 2020 and 2022.**

| Characteristics of individuals surveyed | Survey | | Significant difference (95% CI_b) |
|---|---|---|---|
| | **2020** | **2022** | |
| **Individuals surveyed**<br>N | 15,755 | 15,092 | - |
| **Females**<br>N (%) | 7,869 (50.0) | 7,691 (51.0) | No (-0.3–2.0 pp) |
| **Age**<br>Median (IQR) years | 18 (9–30) | 16 (8–28) | Yes (-2 –-1 years) |
| **Individual was present during the screening interview**<br>N (%) | 13,121 (83.3) | 12,965 (85.9) | Yes (2.1–3.7 pp) |
| **Blindness**<br>N (%) | 445 (2.8) | 435 (2.9) | No (-0.1–0.7 pp) |
| **Age of those blind**<br>Median (IQR) years | 49 (38–61) | 45 (32–56) | Yes (-10 –-3 years) |
| **Skin itching**<br>N (%) | 5,065 (32.2) | 5,290 (35.1) | Yes (1.5–3.6 pp) |

CI_b–Bootstrap confidence interval; IQR–Interquartile range N–Number; pp–Percentage points.

### 3.3 Prevalence of epilepsy in 2020 and 2022

In the surveys conducted in 2020 and 2022, respectively, 815 of 15,755 individuals and 677 of 15,092 individuals with suspected epilepsy were referred to a physician for confirmation (Table 3). Of these, epilepsy was confirmed in 809 (99.3%) individuals in 2020 and 672 (99.3%) individuals in 2022, reflecting a slightly decreasing prevalence trend from 51.4 epilepsy cases per 1,000 persons in 2020 to 44.5 per 1,000 persons in 2022. This decreasing trend was observed in both sexes, with no significant difference in prevalence between males and females in each survey (Table A in S1 Material). The remaining suspected epilepsy cases had other diagnoses, namely a single seizure episode (two in 2020 and three in 2023), psychiatric illness (one in 2020 and one in 2022), severe anaemia (two in 2020), recent severe malaria (one in 2022) and blindness (one in 2020).

Between 2020 and 2022, the prevalence of epilepsy in individuals 20 years old and younger decreased from 40.1 per 1,000 (95% CI_w: 36.2–44.5) to 28.7 per 1,000 (95% CI_w: 25.4–32.5) (Fig 2), representing a decline of 11.4 (95% CI_b: 6.1–16.9) percentage points. However, the

**Table 3. Prevalence of persons with suspected and confirmed epilepsy in 2020 and 2022.**

| Prevalence | Survey | | Significant difference (95% CI_b) |
|---|---|---|---|
| | **2020**<br>**(15,755 individuals)** | **2022**<br>**(15,092 individuals)** | |
| **Suspected lifetime epilepsy** | 815 | 677 | Yes (-1.0 –-0.1 pp) |
| N (prevalence per 1,000 persons, 95% CI_w) | (51.7, 48.4–55.3) | (44.9, 41.6–48.3) | |
| **Confirmed lifetime epilepsy** | 809* | 672** | Yes (-1.0 –-0.1 pp) |
| N (prevalence per 1,000 persons, 95% CI_w) | (51.4, 48.0–54.9) | (44.5, 41.3–48.0) | |

CI_b–Bootstrap confidence interval; N–Number; pp–Percentage points.

* Eleven suspected cases were not confirmed by a clinician, but based on the confirmation rate of 99% in 2020, these individuals were presumed to have confirmed lifetime epilepsy.

** Twenty-two suspected cases were not confirmed by a clinician, but based on the confirmation rate of 99% in 2022, these individuals were presumed to have confirmed lifetime epilepsy.

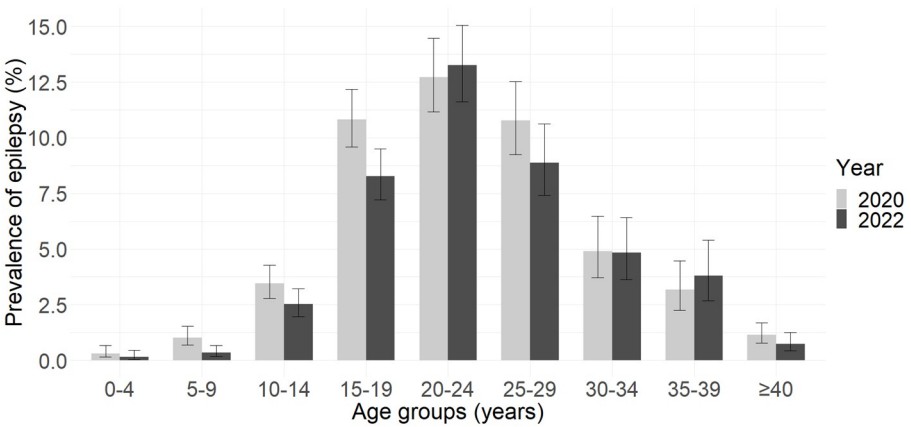

**Fig 2. Prevalence of epilepsy per age group in the population surveyed in 2020 and 2022.**

prevalence remained stable in those older than 20 years, from 63.7 per 1,000 (95% $CI_w$: 58.2–69.7) in 2020 to 63.8 per 1,000 (95% $CI_w$: 57.9–70.2) in 2022.

## 3.4 Sociodemographic and clinical characteristics of PWE in 2022

Both surveys demonstrated a nearly balanced gender distribution among individuals with epilepsy (Table 4), with females comprising 47.4% in 2020 and 47.9% in 2022. The median age of those with epilepsy consistently stood at 20 years. Persons with epilepsy resided in their current site for a median of 20 years, corresponding to their median age, and only a minor fraction were not native (8.4% in 2020 and 8.9% in 2022) with a median residence of 11 years. While most persons with epilepsy had previously taken ivermectin, intake dropped considerably in 2021, with less than one-third (31.7%) taking it, marking a significant sharp decline from the 84.4% reported in 2019. The ethnicities' distribution was consistent among surveys and similar to the ethnic household distribution presented in Table 1.

In 2022, only 23.1% of individuals with epilepsy were receiving antiseizure medication (Table 4), yet a striking 99.4% exhibited active epilepsy (Table 5). Generalised tonic-clonic seizures were the most reported seizures (86.6% in 2022), closely followed by nodding seizures at 37.4%. There was no discernible shift in the types of seizures of the population with epilepsy surveyed between 2020 and 2022. A vast majority experienced seizures at least every month (91.5% in 2022), although fewer reported daily seizures (-11.7%, 95% $CI_b$: -15.7 –- 7.6%). Almost two-thirds of the participants with epilepsy experienced at least one seizure episode in the week preceding the survey.

The median age for the onset of epilepsy was nine years, three (95% $CI_b$: 3–2) years higher than the median age of seven years for the onset of probable NS. Most individuals met the criteria for OAE, namely 73.9% in 2020 and 82.8% in 2022. The detailed reasons for not meeting the OAE criteria are elaborated in Table B in S1 Material. Additionally, a considerable proportion of persons with epilepsy had probable NS (42.0% in 2022).

According to the modified Rankin Scale, a slight decreasing trend in disability was observed, with 15.5% (101) of persons with epilepsy displaying a slight disability and 3.6% (23) having moderate to severe disability in 2022 (Table 6). Furthermore, a lack of full orientation in space, time and person was evident in a fifth of the epileptic population (69 individuals) in 2022, a 5.8 (95% $CI_b$: 0.5–11.0) percentage points rise from 2020. Regarding clinical

**Table 4. Sociodemographic characteristics of persons with epilepsy in 2020 and 2022.**

| Sociodemographic characteristics of individuals with epilepsy | | Survey | | Significant difference (95% CI_b) |
|---|---|---|---|---|
| | | 2020 (798 epilepsy cases) | 2022 (650 epilepsy cases) | |
| **Females** | | 378/797 | 312 | No (-4.7–5.7 pp) |
| N/T (%) | | (47.4) | (48.0) | |
| **Age** | | 20 | 20 | No (-0.78–0.91 years) |
| Median (IQR) years | | (17–25) | (18–25) | |
| **Duration of residence in the village** | | 20 | 20 | No (-0.51–10.0 years) |
| Median (IQR) years | | (15–25) | (16–25) | |
| **Immigrant** | | 67 | 58 | No (-2.2–3.4 pp) |
| N (%) | | (8.4) | (8.9) | |
| **Length of residence of immigrant households in the current site** | | 11 | 11 | No (-4–4 years) |
| Median (IQR) years | | (4–14) | (4–15) | |
| **Ethnic distribution** N (%) | **Jur*** | 747/795 | 623/649 | No (-0.5–4.1 pp) |
| | | (94.0) | (96.0) | |
| | **Muro** | 41/795 | 18/649 | No (-3.6–0.1 pp) |
| | | (5.1) | (2.8) | |
| | **Dinka** | 4/795 | 3/649 | No (-0.8–0.7 pp) |
| | | (0.5) | (0.4) | |
| | **Other/unidentified**** | 3/795 | 5/649 | No (-0.4–1.2 pp) |
| | | (0.4) | (0.8) | |
| **Ivermectin intake in the past** N/T (%) | **In the past** | 661/756 | 351/630 | Yes (-32.5 – -23.3 pp) |
| | | (87.4) | (55.7) | |
| | **In the year before the survey** | 645/764 | 205/630 | Yes (-49.2 – -39.9 pp) |
| | | (84.4) | (32.5) | |
| **Under antiseizure medication** N (%) | **Yes, currently** | 154/781 | 149/644 | No (-0.9–7.7 pp) |
| | | (19.7) | (23.1) | |
| | **Yes, in the past** | 488/781 | 266/644 | Yes (-26.2 – -16.1 pp) |
| | | (62.5) | (41.3) | |
| | **Never** | 139/781 | 229/644 | Yes (13.2–22.3 pp) |
| | | (17.8) | (35.6) | |

N–Number; pp–Percentage points; T–Total, if different from the overall epilepsy cases confirmed by a clinician; IQR–Interquartile range.

* Sub-ethnicities included in the Jur ethnic group were Jur Modo/Beli (2,041), Nyamusa/Nyamusa Molo/Kpelikiri (74), Wira (52), Lali (2), Lori (1) and Sopi (1).

** Such as Madi (4), Baya (3), Paliki (2) and Nuba (1).

manifestations possibly related to onchocerciasis, over half the population with epilepsy had skin manifestations (e.g. 66.7% in 2022) and 2–3% experienced blindness in at least one eye. Those with epilepsy and blindness tended to be younger than their blind counterparts without epilepsy in both surveys—2020 (p = 0.010) and 2022 (p<0.001).

### 3.5 Ivermectin coverage in 2019 and 2021

During the 2020 survey, 9,896 (62.8%) participants reported taking ivermectin in 2019. However, only 3,629 (24.1%) participants reported taking ivermectin in 2021 (data from the 2022 survey; see Table 7). This reflects a significant reduction in coverage of 38.8 (95% CI_w: 37.8–39.8) percentage points. This decline was consistent across all age groups, with particularly lower coverage observed within the younger age groups.

**Table 5. Epilepsy clinical characteristics of persons with epilepsy in 2020 and 2022.**

| Epilepsy clinical characteristics of individuals with epilepsy | | Survey | | Significant difference (95% CI_b) |
|---|---|---|---|---|
| | | **2020 (798 epilepsy cases)** | **2022 (650 epilepsy cases)** | |
| Current types of seizures N (%) | Generalised tonic-clonic | 641 (80.3) | 563 (86.6) | Yes (0.1–8.6 pp) |
| | Atonic (drop attacks) | 79 (9.9) | 40 (6.2) | Yes (-4.7 –-0.7 pp) |
| | Absences | 79 (9.9) | 57 (8.8) | No (-3.0–1.3 pp) |
| | Focal motor with conserved consciousness | 1 (0.1) | 5 (0.7) | No (0.0–1.0 pp) |
| | Focal motor with altered consciousness | 4 (0.5) | 7 (1.1) | No (-0.2–1.1 pp) |
| | Nodding seizures | 317 (39.7) | 243 (37.4) | No (-5.7–2.2 pp) |
| Frequency of seizures N/T (%) | Daily seizures | 210/782 (26.9) | 99 (15.2) | Yes (-15.7 –-7.6 pp) |
| | Weekly seizures | 191/782 (24.4) | 205 (31.5) | Yes (2.5–11.7 pp) |
| | Monthly seizures | 342/782 (43.7) | 291 (44.8) | No (-4.1–6.2 pp) |
| | Yearly seizures | 39/782 (5.0) | 55 (8.5) | Yes (0.9–6.1 pp) |
| Seizures experienced the week preceding the survey | | 492/761 (64.7) | 418/644 (64.9) | Yes (-3.9–6.2 pp) |
| Age at onset of seizures* Median (IQR) years | | 9* (6–13) | 9* (7–12) | No (-1–1 years) |
| Active epilepsy N (%, 95% CI) | | 788 (98.8, 97.6–99.4) | 646 (99.4, 98.3–99.8) | No (-0.4–1.6 pp) |
| Onchocerciasis-associated epilepsy† N (%, 95% CI) | | 541 (73.9, 70.5–77.0) | 515 (82.8, 79.6–85.6) | Yes (4.5–13.3 pp) |
| Probable nodding syndrome‡ N (%, 95% CI) | | 354 (44.4, 40.9–47.9) | 273 (42.0, 38.2–45.9) | No (-7.5–2.7 pp) |
| Age at onset of probable nodding syndrome§ Median (IQR) years | | 7 (5–10) | 7 (5–10) | No (-1–1 years) |

CI_b–Bootstrap confidence interval N–Number; pp–Percentage points; T–Total, if different from the overall epilepsy cases confirmed by a clinician; IQR–Interquartile range.

* Data were only collected from 706 individuals in 2020 and 613 individuals in 2022.

† Sixty-six persons with epilepsy in 2020 and 28 in 2022 met all the criteria for onchocerciasis-associated epilepsy (OAE), except for the age of seizure onset, which was unknown. These cases were, therefore, excluded from the calculations of the OAE variable.

‡ In addition to individuals experiencing "current nodding seizures", the category of "probable nodding syndrome" also accounts for individuals with epilepsy who had nodding seizures in the past. In 2022, 218 (80%) of the individuals with probable nodding syndrome had other types of seizures besides nodding seizures. Similarly, in 2020, 264 out of 342 individuals (77%) with probable nodding syndrome experienced other types of seizures. This co-occurrence of nodding and other seizures is denoted as nodding syndrome plus and is within the spectrum of onchocerciasis-associated epilepsy [8].

§ Data were only collected from 252 individuals with probable nodding syndrome in 2020 and 248 in 2022.

### 3.6 Incidence of epilepsy, including NS, before and during annual CDTI

Between 2013–2015 and 2019–2021, the incidence of epilepsy gradually decreased from 326.5 (95% CI_w: 266.8–399.1) cases per 100,000 person-years to 96.6 (95% CI_w: 65.5–141.7) per 100,000 person-years (Table 8). This represents a significant decline of 70.4% (95% CI_w: 56.4–81.7%). The incidence of NS mirrored the incidence of overall epilepsy, decreasing from 151.7 (95% CI_w: 112.7–203.4) per 100,000 person-years in 2013–2015 to 27.0 (95% CI_w: 12.5–55.5) per 100,000 person-years in 2019–2021.

## 4. Discussion

Cumulative ivermectin distribution, even with suboptimal coverage, was followed by a marked reduction in the overall incidence of epilepsy cases per 100,000 person-years from 326.5 (95%

**Table 6. Associated clinical features of persons with epilepsy in 2020 and 2022.**

| Other clinical characteristics of individuals with epilepsy | | Survey | | Significant difference (95% CI$_b$) |
|---|---|---|---|---|
| | | 2020 (798 epilepsy cases) | 2022 (650 epilepsy cases) | |
| **Blindness in one or two eyes** | | 16/585 | 8/337 | No (-2.4–1.8 pp) |
| N/T (%) | | (2.7) | (2.4) | |
| **Age of those blind** | | 33.5 | 22.0 | Yes (-6.9 - -2.0 years) |
| Median (IQR) years | | (15.3–43.8) | (20.8–28.5) | |
| **Onchocerciasis-related skin manifestations** | | 314/584 | 224/336 | Yes (6.3–19.2 pp) |
| N (%) | | (53.8) | (66.7) | |
| **Nakalanga features** | | 147/556 | 54/304 | Yes (-14.3–3.0 pp) |
| N (%) | | (26.4) | (17.8) | |
| **Not oriented (in space/time/person)** | | 86/584 | 69/337 | Yes (0.5–11.0 pp) |
| N (%) | | (14.7) | (20.5) | |
| **Modified Rankin Scale for neurologic disability** N (%) | **No significant disability** | 425/572 | 526/650 | Yes (0.8–8.5 pp) |
| | | (74.3) | (80.9) | |
| | **Slight to moderate** | 125/572 | 101/650 | Yes (-10.7 –-1.8 pp) |
| | **disability** | (21.9) | (15.5) | |
| | **Moderately severe to severe disability** | 22/572 | 23/650 | No (-2.5–1.8 pp) |
| | | (3.9) | (3.6) | |

CI$_b$–Bootstrap confidence interval; N–Number; IQR–Interquartile range; pp–Percentage points; T–Total.

CI: 266.8–399.1) in 2013–2015 to 96.6 (95% CI: 65.5–141.7) in 2019–2021. In particular, the incidence of probable NS per 100,000 person-years also decreased from 151.7 (95% CI: 112.7–203.4) to 27.0 (95% CI: 12.5–55.5) cases. These declines are consistent with trends observed in other areas highly endemic for onchocerciasis following the introduction or strengthening of control interventions, such as in Maridi County in South Sudan [14] and the Mahenge area in Tanzania [47].

In Maridi County, annual ivermectin distribution achieving 40–60% therapeutic coverage was followed by a significant reduction in the incidence of overall epilepsy from approximately 350 to 165 cases per 100,000 person-years [14]. Similarly, the incidence of probable NS significantly declined from 155 to 40 cases per 100,000 person-years. Further intensification of onchocerciasis control measures in Maridi, including a shift to biannual (twice-yearly) CDTI

**Table 7. Ivermectin coverage in 2019 and 2021.**

| Ivermectin coverage | | 2019 N/T (%, 95% CI$_w$) | 2021 N/T (%, 95% CI$_w$) | Difference pp (95% CI$_w$) |
|---|---|---|---|---|
| **Overall (therapeutic coverage)** | | 9,896/15,748 (62.8, 62.1–63.6) | 3,629/15,092 (24.1, 23.4–24.7) | -38.8 (-37.8 –-39.8) |
| **Per age group** | **0–4 years*** | 0/1,924 | 0/1,957 | - |
| | **5–9 years** | 1,329/2,246 (59.2, 57.1–61.2) | 414 /2,315 (17.9, 16.4–19.5) | -41.3 (-38.7 –-43.9) |
| | **10–19 years** | 3,227/4,549 (70.9, 69.6–72.3) | 1,151 /4,641 (24.8, 23.6–26.1) | -46.1 (-44.3 –-48.0) |
| | **>20 years** | 5,340/7,029 (76.0, 75.0–77.0) | 2,064 /6,179 (33.4, 32.2–34.6) | -42.6 (-41.0 –-44.1) |

CI$_w$–Wilson score confidence interval; N–Number; pp–Percentage points; T–Total.

* The safety of oral ivermectin in children under five years of age has not been established and is therefore not delivered to this age group.

**Table 8. Overall epilepsy and probable nodding syndrome incidence from 2013 to 2021.**

| Time periods | Overall epilepsy incidence | | Probable NS incidence | |
|---|---|---|---|---|
| | n; T | n/100,000 (95% CI$_w$) PY | n; T | n/100,000 (95% CI$_w$) PY |
| July 2013 –June 2015 Average (S1) | 99; 30,320 | 326.5 (266.8–399.1) | 47; 30,990 | 151.7 (112.7–203.4) |
| July 2015 –June 2017 Average (S1) | 66; 30,122 | 219.1 (170.8–280.5) | 20; 30,896 | 64.7 (40.6–101.9) |
| July 2017 –June 2019 Average (S1) | 41; 29,990 | 136.7 (99.4–187.3) | 23; 30,856 | 74.5 (48.4–113.8) |
| July 2019 –June 2021 Average (S2) | 28; 28,974 | 96.6 (65.5–141.7) | 8; 29,656 | 27.0 (12.5–55.5) |

CDTI–Community-directed treatment with ivermectin; CI$_w$–Wilson score confidence interval; NS–Nodding syndrome; n–number of new-onset epilepsy; PY–Person-years; S2 Dataset–Data from 2022 survey; S1 Dataset–Data from 2020 survey; T- Total number of person-years at risk of developing epilepsy.

and the adoption of a community-based Slash & Clear vector control method, was succeeded by a further decrease in epilepsy incidence rates to numbers comparable with those reported in sub-Saharan Africa regions not heavily affected by onchocerciasis [54].

In the Mahenge area, the transition from suboptimal annual CDTI to biannual CDTI with optimal 80% coverage was linked to a significant reduction in overall epilepsy incidence from approximately 180 to 45 cases per 100,000 person-years [47]. The decrease in probable NS incidence, from 20 to 5 cases per 100,000 person-years, was not statistically significant, as the NS incidence was already relatively low during suboptimal annual CDTI. As a severe type of OAE, NS is believed to be associated with high *O. volvulus* microfilarial loads [11], which may be considerably suppressed with a suboptimal CDTI coverage of 60% [55,56]. This hypothesis is also supported by the rapid decline in the incidence of probable NS in both Mvolo and Maridi Counties following suboptimal ivermectin distribution (Table C in S1 Material) [47].

The observed decline in epilepsy incidence in Mvolo began around 2016, as illustrated in Fig A in S1 Material. This downward trend aligns with the phased implementation of the CDTI programme, initiated in the early 2000s. While the programme initially achieved very low coverage, it progressively expanded over time, albeit remaining below the optimal threshold recommended by the World Health Organization for the elimination of onchocerciasis transmission [39]. The reduction in epilepsy incidence became evident approximately a decade after the CDTI programme fully covered Mvolo. This delay aligns with the time required for ivermectin to effectively reduce the microfilarial load among the age group at risk for OAE, particularly in the context of Mvolo's suboptimal annual CDTI coverage and high baseline endemicity [26,55–57].

Moreover, the suppression of microfilariae production in the adult population, which largely contributes to the transmission of *O. volvulus*, indirectly reduces the exposure of young children to onchocerciasis. It is important to note that the younger age groups, either excluded from treatment until the age of five years or receiving it at lower rates (five to 19 years), were expected to hold considerable microfilarial loads [57]. Evidence of high ongoing onchocerciasis exposure among younger demographics in Mvolo was shown in 2020, with a substantial Ov16 seroprevalence among children under ten (27.3%, 95% CI: 20.8–34.9%) [41]. Consequently, a noticeable impact of CDTI on the onchocerciasis burden of younger demographics is expected to manifest several years after its initiation. This underscores the observed decrease in epilepsy incidence approximately a decade after the start of extensive CDTI. Since the risk of developing OAE is believed to be related to the skin microfilarial load of three- to 18-year-olds [11], the potentially pivotal role of ivermectin in reducing this load, especially in highly endemic areas like Mvolo, cannot be overstated.

While the prevalence of epilepsy in Mvolo remains high at 4–5%, there was a notable reduction in prevalence among those under 20 years of age by 11.4 (95% CI: 6.1–16.9) percentage

points from 2020 to 2022. This reduction supports the observed decline in epilepsy incidence, which in endemic areas is predicted to predominantly occur in children and adolescents when onchocerciasis elimination measures are introduced or strengthened [15]. A similar result was described in Maridi County, with a decrease in epilepsy prevalence of 16.4 (95% CI: 11.3–21.5%) percentage points for the same age group between 2018 and 2022 [14]. In contrast, epilepsy prevalence in individuals aged over 20 years remained unchanged in Mvolo, while Maridi saw a significant increase of 16.3%. Another relevant metric is the proportion of persons with epilepsy "not oriented in time, space and person", which slightly increased in Mvolo (from 14.7% to 20.5%), while significantly decreasing in Maridi (from 21.3% to 9.7%). This difference is likely explained by Maridi's longer study duration and lower epilepsy-related mortality [58] due to the establishment of epilepsy clinics in 2020, where antiseizure medication is provided free of charge [14]. Whereas the reported uptake of antiseizure medications significantly increased in Maridi from 51.4% in 2018 to 91.1% in 2022, it was persistently low in Mvolo (~20%) in 2020 and 2022. These findings emphasise the need to decrease the epilepsy treatment gap in Mvolo and across sub-Saharan Africa [59,60].

Despite a significant reduction in epilepsy incidence in Mvolo, ivermectin uptake was substantially below the recommended 80% therapeutic coverage for onchocerciasis elimination [24], achieving only 62.8% (95% CI: 62.1–63.6%) in 2019. The COVID-19 pandemic led to the suspension of CDTI across South Sudan in 2020 and a subsequent dramatic decrease in ivermectin uptake to 24.1% (95% CI: 23.4–24.7%) in 2021. While long-established optimal CDTI programmes might endure a one-year interruption without a substantial rise in community microfilarial loads [57,61], programmes such as the one in Mvolo are more vulnerable. The latter has been illustrated by mathematical modelling [57], which suggests that children are at high risk of increasing microfilarial loads. Added to this, as previously described, ivermectin uptake in Mvolo was consistently lower among younger age groups, especially those aged five to ten years, who are most susceptible to developing OAE [8]. Understanding and addressing the underlying reasons for the low ivermectin uptake is therefore vital. Strategies could include reducing ivermectin misconceptions [62], raising OAE awareness [47] and adapting CDTI strategies to local contexts, such as considering seasonal farming migrations [63]. This should be a priority not only for Mvolo County but also across the onchocerciasis-endemic regions of South Sudan, given that all states in the country are endemic [64] and have reported less than five rounds of optimal annual CDTI [24].

In 2016, 20% (12,114/60,571) of the population in Mvolo was under five years old [19], thereby excluded from ivermectin treatment. When also factoring in the proportion of women ineligible for treatment due to pregnancy or early lactation, achieving the optimal 80% CDTI coverage in Mvolo may not be feasible, even if all the other conditions for an effective CDTI are met. Given the pre-control holoendemicity of Mvolo, with around 90% onchocerciasis prevalence, it is unlikely that elimination targets would be achieved solely under annual CDTI [65,66]; therefore, alternative treatment strategies could be necessary. The high Ov16 seroprevalence found among young children in Mvolo in 2020 further indicates ongoing *O. volvulus* transmission under the current annual CDTI strategy [41].

Alternative strategies for onchocerciasis elimination could include biannual CDTI and complementary vector control measures. The latter is particularly promising in South Sudan and notably in Mvolo, where blackfly populations are isolated by considerable distances between river basins, which minimises the risk of reinfestation [67]. The effectiveness of focal vector control implementation in South Sudan, demonstrated by an 80% reduction in the annual transmission potential of onchocerciasis in Bahr el Gazal State [68], supports the feasibility of such an approach. Informed by the current and previous studies in Western Equatoria [14,48], Mvolo and neighbouring counties are now targeted for biannual CDTI and, wherever

possible, a community-based Slash & Clear vector control method [69]. The latter method has been showing promising results in Maridi County [14] and Uganda [70]; however, its impact on areas with multiple breeding sites remains to be determined [69]. Other vector control strategies might be more challenging to implement, not only due to the rainy season [20] but also due to their substantial costs, especially considering the poverty in the area. Thus, increasing the frequency of CDTI may be the most practical approach to potentially preventing the onset of OAE.

Between 2020 and 2022, there was a significant increase in epilepsy cases meeting the OAE criteria, from 73.9% to 82.8%. This can be largely attributed to the substantially lower proportion of persons with epilepsy reporting a history of severe malaria in 2022 (S1 Material). Malaria is a leading cause of febrile seizures in children up to the age of six years and may pose a challenge in the diagnosis of epilepsy [71]. Nevertheless, its impact on the findings of the current study is likely limited, as the median age of onset of seizures in Mvolo was ten years (IQR: 6–12 years), and most persons with active epilepsy were young adults, with a median age of 20 years (IQR: 18–25 years).

A strength of the study was the consistency in the methodology and the retention of some research team members across surveys in Mvolo, Maridi and Mahenge, contributing to the reliability of the findings and integrated analysis. For instance, the substantial decline in epilepsy incidence started between 2015 and 2017 in the geographically and administratively distinct Mvolo and Maridi Counties, coinciding with the reintroduction of ivermectin distribution in both areas. This temporal alignment reinforces the observed association between the uptake of ivermectin and the observed reduction in epilepsy incidence. Moreover, the use of an adjusted bootstrap method to calculate confidence intervals, accounting for the 33% overlap found in households surveyed across the two years, further enhances this reliability. This adjustment acknowledges the non-independence of samples due to repeated household participation in successive surveys, providing a more realistic representation of our data. Additionally, a sensitivity analysis was conducted using the Wilson score interval with Yate's continuity correction on all presented bootstrap 95% CIs to ensure the robustness of the bootstrap CIs to alternative methods and potential violations of normality assumptions. Consistent results between the CIs obtained from the bootstrap method and the Wilson score provided further confidence in the findings.

This study faced several limitations. For instance, the absence of a surveillance system for new epilepsy cases during the two-year study period meant that incidence data relied solely on interviews conducted during the two cross-sectional surveys conducted. To address the potential bias of overlooking newly developed epilepsy cases in the incidence calculations, the 2022 survey was used retrospectively to calculate the incidence for the year preceding the 2020 survey, and newly diagnosed epilepsy cases reported between 2022 and 2021 were excluded from the incidence rate calculations to prevent underestimation. However, there remains a possibility of recall bias when retrospectively calculating the epilepsy incidence and ivermectin coverage. The former was mitigated by calculating the incidence every two years. Concerning the latter, while there is a potential for recall bias as coverage data were assessed eight and ten months after the last treatment round in 2019 and 2021, respectively, the high endemicity of onchocerciasis in Mvolo, the community's awareness of the condition, which was targeted by an awareness campaign in 2020 [41], and the need for preventive treatment may help minimise this bias. Moreover, directly asking the population about their intake of ivermectin provides additional insights besides the coverage reported by the community drug distributors, such as accounting for individuals who received ivermectin but opted not to take it (e.g. reason for non-adherence) [49,73,74], and households inadvertently overlooked during the CDTI rounds.

The cross-sectional surveys, while pragmatic, lack the robustness of a cohort study, with potential variances in the households surveyed in 2020 and 2022 possibly affecting the reliability of the results. However, the consistent epilepsy prevalence among individuals over 20 years old in Mvolo and the concordance of epilepsy types and demographics suggest that very similar populations were sampled in each survey. Another concern is the potential for undocumented population movements due to recent floods, which may have caused families to relocate, impacting the demographic stability of the county. Nevertheless, the primary income sources, farming and fishing, which are activities at high risk for blackfly bites, remained largely unchanged and were reported by 95% of households, indicating continued exposure to onchocerciasis vectors.

Another limitation is that the study could not employ epilepsy gold-standard diagnostic methods such as electroencephalogram (EEG) or magnetic resonance imaging (MRI). However, the diagnostic methods used have been validated by the ILAE [74] and have consistently yielded similar results of epilepsy reduction following the implementation of onchocerciasis control strategies in endemic areas [11,14,47]. Lastly, although there was no testing for *Taenia solium* infection, pig rearing was reported by less than 1.0% of households. Hence, the high epilepsy incidence and prevalence in Mvolo are very unlikely to be explained by neurocysticercosis.

## 5. Conclusions

In Mvolo, the sustained implementation of CDTI, despite the challenges of achieving optimal coverage, has been followed by a significant reduction in the incidence of epilepsy. This study further aligns with the broader understanding that onchocerciasis elimination programmes can successfully reduce the incidence of epilepsy in areas with high *O. volvulus* transmission and epilepsy burden. Hence, there is a pressing need to strengthen onchocerciasis elimination efforts in Mvolo and across South Sudan. Concurrently, it is essential to enhance access to epilepsy treatment services to mitigate epilepsy-related morbidity and mortality, especially in areas of onchocerciasis hyperendemicity that have experienced suboptimal onchocerciasis elimination programmes.

### Ethical approval

Ethical approval was obtained from the ethics committee of the Ministry of Health of South Sudan (January 2018, MOH/ERB50/2019) and from the ethics committee of the Antwerp University Hospital, Belgium (April 2019, B300201940004).

### Informed consent statement

The aims and procedures of the study were explained to all participants in the language of their choice, and signed or thumb-printed informed consent was obtained from participants, parents or caregivers, and assent was obtained from adolescents (aged 12–18 years).

### Supporting information

**S1 Dataset. Persons with epilepsy data 2020.**
(XLSX)

**S2 Dataset. Persons with epilepsy data 2022.**
(XLSX)

**S3 Dataset. Household data 2020.**
(CSV)

**S4 Dataset. Household data 2022.**
(CSV)

**S5 Dataset. Individual data 2020.**
(CSV)

**S6 Dataset. Individual data 2022.**
(CSV)

**S1 Material.** Questionnaire A–Epilepsy screening questionnaire delivered to every element of each household to identify persons with suspected epilepsy. Questionnaire B–Neurology questionnaire delivered to the persons with suspected epilepsy to confirm or reject the epilepsy diagnosis. Fig A–Incidence of epilepsy trends over time in Mvolo County. Table A–Gender-specific epilepsy prevalence in 2020 and 2022 in Mvolo County. Table B—Obvious causes of epilepsy. Table C–Longitudinal decrease in the incidence of overall epilepsy and probable nodding syndrome (NS) after reintroducing ivermectin distribution in Maridi and Mvolo Counties.
(PDF)

## Acknowledgments

We thank the population of Mvolo and the local research assistants for their participation and Dr Alfred Dusabimana for his support.

## Author Contributions

**Conceptualization:** Luís-Jorge Amaral, Stephen Raimon Jada, Jane Y. Carter, Thomson Lakwo, Robert Colebunders.

**Data curation:** Luís-Jorge Amaral, Stephen Raimon Jada.

**Formal analysis:** Luís-Jorge Amaral, Aimee Kemayou Ndjanfa, Joseph N Siewe Fodjo.

**Funding acquisition:** Luís-Jorge Amaral, Joseph N Siewe Fodjo, Robert Colebunders.

**Investigation:** Stephen Raimon Jada, Gasim Abd-Elfarag, Samuel Okaro, Thomson Lakwo.

**Methodology:** Luís-Jorge Amaral, Stephen Raimon Jada, Robert Colebunders.

**Project administration:** Stephen Raimon Jada.

**Resources:** Robert Colebunders.

**Software:** Luís-Jorge Amaral.

**Supervision:** Robert Colebunders.

**Validation:** Luís-Jorge Amaral, Stephen Raimon Jada.

**Visualization:** Luís-Jorge Amaral, Thomson Lakwo.

**Writing – original draft:** Luís-Jorge Amaral, Stephen Raimon Jada, Aimee Kemayou Ndjanfa, Joseph N Siewe Fodjo, Robert Colebunders.

**Writing – review & editing:** Luís-Jorge Amaral, Stephen Raimon Jada, Aimee Kemayou Ndjanfa, Jane Y. Carter, Gasim Abd-Elfarag, Makoy Yibi Logora, Yak Yak Bol, Thomson Lakwo, Joseph N Siewe Fodjo, Robert Colebunders.

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
