## [Decision Letter · Decision Letter 0]

1 Feb 2024

Dear Mr Amaral,

Thank you very much for submitting your manuscript "Impact of annual community-directed treatment with ivermectin on the incidence of epilepsy in Mvolo, a two-year prospective study" for consideration at PLOS Neglected Tropical Diseases. As with all papers reviewed by the journal, your manuscript was reviewed by members of the editorial board and by several independent reviewers. In light of the reviews (below this email), we would like to invite the resubmission of a significantly-revised version that takes into account the reviewers' comments. 

We cannot make any decision about publication until we have seen the revised manuscript and your response to the reviewers' comments. Your revised manuscript is also likely to be sent to reviewers for further evaluation.

Sincerely,

Richard Reithinger

Academic Editor

Uriel Koziol

Section Editor

Reviewer's Responses to Questions

**Key Review Criteria Required for Acceptance?**

**Methods**

-Are the objectives of the study clearly articulated with a clear testable hypothesis stated?

-Is the study design appropriate to address the stated objectives?

-Is the population clearly described and appropriate for the hypothesis being tested?

-Is the sample size sufficient to ensure adequate power to address the hypothesis being tested?

-Were correct statistical analysis used to support conclusions?

-Are there concerns about ethical or regulatory requirements being met?

Reviewer #1: The methods need some more details and clarification.

Page 3: What does the term, “reduced periods of consciousness” mean? Does it refer to duration of impairment of consciousness?

Page 8, paragraph 1: The sentence refers to “five epilepsy screening questions”. But no details are provided. What were these questions?

Page 10: It is understandable that household surveys in this region of the world are challenging and particularly for making medical diagnoses in the field. But it would be useful to know how specific are the criteria being used for determining these clinical manifestations. For example, many exematous skin conditions could present with nodules, pruritis and scars. Similarly the assessment of cognitive function is limited to orientation. This would be considered highly inadequate for assessment of cognitive function. Nakalanga features are defined as thoracic or spine abnormalities with growth retardation. This also seems very non-specific for many medical conditions can cause such a phenoypye.

Reviewer #2: Accept

Reviewer #3: The study design is strengthened by the clinically-confirmed diagnosis of epilepsy, including pNS. The study design was also clear, being described as two cross-sectional surveys. 

Page 8: 

The methodology is incomplete, as there is no description of the ivermectin regimen, such as what months the participants took the treatment or the specific dosage of treatment. Additionally, no details are provided on when the surveys were conducted after treatment. How many weeks or months after ivermectin intake were the participants surveyed? If it was long after, there could be an issue of recall bias. 

Page 10, paragraph 1: 

Onchocerciasis-related blindness was clinically defined, but similar blindness can be caused by trachoma, a public health issue also endemic in South Sudan, and vitamin A deficiency. Thus, there is a caveat to this definition, and the limitation of the clinical diagnosis should be discussed later in the manuscript as well. 

Page 10, paragraph 1: 

Cognitive impairment was defined as "when an individual is not fully oriented in space, time, and person." However, this symptom can be caused by a number of things that are not included in the manuscript's research. Therefore, this variable is not well defined in relation to onchocerciasis. 

Page 10, paragraph 2: 

The data analysis involved a "bootstrapping method" and the "Bootstrap 95% Confidence Interval." Given that this is an uncommonly used methodology, it should be at least referenced by an outside source in this section.

**Results**

-Does the analysis presented match the analysis plan?

-Are the results clearly and completely presented?

-Are the figures (Tables, Images) of sufficient quality for clarity?

Reviewer #1: Data analysis: No P values have been calculated for any of the observations. This is particularly important to substantiate any of the claim for decrease in the incidence or prevalence of NS or epilepsy in the population over time.

Page 15: What criteria were used to diagnose “onchocerca related blindness”? Does this population also have other causes of blindness such as vitamin A deficiency or trachoma etc.

Table 4: Please include dosage and frequency of administration of ivermectin.

Table 5: How many patients with Nodding Syndrome also had other types of seizures? This could be an interesting population to study. It might be important to know if ivermectin had any impact of these different types of seizures in the same individual. Along the same lines, once NS has developed does ivermectin have any effect on the seizures?

Table 5: It gives the frequency of seizures. But often they occur in clusters and not neatly in regular intervals of daily, weekly, etc. This information needs to be revised.

Reviewer #2: Accept

Reviewer #3: All of the results were presented in tables and the abbreviations were explained below the relevant tables. 

Data analysis: 

Again, how does the Bootstrap 95% Confidence Interval relate to statistical significance? How can one tell the confidence interval is statistically significant without the calculation of p values?

Page 12, paragraph 1; page 16, paragraph 1: 

The terms "percent" and "percentage points" are used interchangeably when referring to a change in the data points for the two surveys. This is incorrect, as only percentage points refers to the difference between percentages.

**Conclusions**

-Are the conclusions supported by the data presented?

-Are the limitations of analysis clearly described?

-Do the authors discuss how these data can be helpful to advance our understanding of the topic under study?

-Is public health relevance addressed?

Reviewer #1: Page 26: Discussion: In the section on vector control strategies it would be useful to also discuss if besides humans if there are other animal reservoirs for onchocercha.

Discussion: It might be useful to speculate as to why partial protection with invermectin continues to have a protective response against the development of NS. 

Are there other parasitic infections in the population that could be impacted by the use of ivermection. Could they play a role in the clinical syndrome?

It would be helpful to discuss the potential pathophysiological mechanisms by which onchocercha infection might cause nodding syndrome. This would help strengthen their conclusions and observations.

Reviewer #2: Accept

Reviewer #3: Discussion: 

There are some limitations missing from the discussion section. While the implementation of CDTI was concluded to be followed by a reduction in epilepsy, it is not discussed why the results section included data (table 8) showing epilepsy on the decline since 2013. Additionally, the manuscript infers implementation of ivermectin only from 2019-2021. Thus, is there a possibility the decrease between 2020-2022 is due to the continuation of a previous trend, confounding variables, or other standard of care measures rather than ivermectin? This limitation regarding ivermectin-related reduction in epilepsy is especially concerning given the suboptimal coverage by the second survey. Or, can the author confirm and specify if rounds of ivermectin treatment were distributed from 2013 onwards, possibly contributing to the trend of decreased epilepsy incidence. 

Page 25, paragraph 1: 

It would be useful to emphasize the importance of not only strengthening access to ivermectin, but also increasing treatment coverage specifically towards the younger, more at risk age group. This is especially important as the younger age group had decreased CDTI compared to the older participant group.

**Editorial and Data Presentation Modifications?**

Reviewer #1: (No Response)

Reviewer #2: Minor Revision ; an explanation on why mosquito nets were considered in the data collection would provide clarity of the significance of bed net use in Onchocerciasis and OAE.

Reviewer #3: (No Response)

**Summary and General Comments**

Reviewer #1: This is an interesting study that shows that the use of an antiparasitic agent can have a dramatic effect of the incidence of epilepsy in a population with a high incidence of onchocerca infection. The observations are of public health importance and could have important socio-economic ramifications in these populations.

Reviewer #2: Great work adding to the evidence of OAE .

Reviewer #3: This study is important in emphasizing a relation between onchocerciasis and epilepsy, as epilepsy incidence may also improve with the use of ivermectin in endemic regions. This link is important for the public health field, as increased CDTI could lead to health benefits among these populations. However, there needs to be some more thought to the limitation section, specifically regarding the previously declining epilepsy trend.
---

## [Editor Report · Decision Letter 1]

8 Mar 2024

Dear Mr Amaral,

We are pleased to inform you that your manuscript 'Impact of annual community-directed treatment with ivermectin on the incidence of epilepsy in Mvolo, a two-year prospective study' has been provisionally accepted for publication in PLOS Neglected Tropical Diseases.

Best regards,

Richard Reithinger

Academic Editor

Uriel Koziol

Section Editor

Thank you for comprehensively addressing the reviewers' comments and concerns.

---

## [Editor Report · Acceptance letter]

18 Mar 2024

Dear Mr Amaral,

We are delighted to inform you that your manuscript, "Impact of annual community-directed treatment with ivermectin on the incidence of epilepsy in Mvolo, a two-year prospective study," has been formally accepted for publication in PLOS Neglected Tropical Diseases.

Best regards,

Shaden Kamhawi

co-Editor-in-Chief

Paul Brindley

co-Editor-in-Chief
